# Automatic Classification System of Arrhythmias Using 12-Lead ECGs with a Deep Neural Network Based on an Attention Mechanism †

**Dengao Li** [1,2,*] **, Hang Wu** [3] **, Jumin Zhao** [2,3] **, Ye Tao** [3] **and Jian Fu** [1]

[1] College of Data Science, Taiyuan University of Technology, Jinzhong 030600, China; fujian1027@link.tyut.edu.cn

[2] Technology Research Center of Spatial Information Network Engineering of Shanxi, Jinzhong 030600, China; zhaojumin@tyut.edu.cn

[3] College of Information and Computer, Taiyuan University of Technology, Jinzhong 030600, China; wuxing0238@link.tyut.edu.cn (H.W.); taoye0237@link.tyut.edu.cn (Y.T.)

* Correspondence: lidengao@tyut.edu.cn; Tel.: +86-138-0349-1195

† This article belongs to the Special Issue Computer and Engineer Science and Symmetry.

**Abstract:** Nowadays, a series of social problems caused by cardiovascular diseases are becoming increasingly serious. Accurate and efficient classification of arrhythmias according to an electrocardiogram is of positive significance for improving the health status of people all over the world. In this paper, a new neural network structure based on the most common 12-lead electrocardiograms was proposed to realize the classification of nine arrhythmias, which consists of Inception and GRU (Gated Recurrent Units) primarily. Moreover, a new attention mechanism is added to the model, which makes sense for data symmetry. The average F1 score obtained from three different test sets was over 0.886 and the highest was 0.919. The accuracy, sensitivity, and specificity obtained from the PhysioNet public database were 0.928, 0.901, and 0.984, respectively. As a whole, this deep neural network performed well in the multi-label classification of 12-lead ECG signals and showed better stability than other methods in the case of more test samples.

**Keywords:** 12-leads electrocardiogram; deep neural network; arrhythmia classification system; attention mechanism

## 1. Introduction

Electrocardiograms (ECG), as a technical means to record the changes of electrical activity generated by each cardiac cycle, have made outstanding contributions in clinical medicine in the past, especially in the diagnosis of arrhythmia and myocardial infarction [1,2]. It is difficult for doctors to make efficient and accurate diagnoses in the face of tens of thousands of ECG records from different individuals. In addition, there are a lot of noise interferences in the originally collected ECG signals, and the non-obvious potential deviation of special nodes also causes great trouble for cardiologists. With the rapid development of computer-aided diagnosis technology, most commercial ECG machines often have a built-in arrhythmia automatic diagnosis algorithm, but its high misdiagnosis rate is unacceptable [3]. In recent decades, researchers have tried to incorporate medical theory into the automated computer analysis of electrocardiograms for the purpose of accurate diagnosis.

As the most commonly used auxiliary diagnostic method of heart disease, ECG contains abundant cardiac beat information and clinical features. Classification of arrhythmias based on ECG signals is of great significance for effective diagnosis, treatment, and early warning of various cardiovascular diseases. Most classical ECG classification methods are based on single-lead methods, which are

often accompanied by tedious operations such as filtering, waveform feature extraction, and R point positioning before classifier construction [4–6]. Machine learning methods have shown great potential in solving tasks in numerous academic and industrial fields [7,8]. The achievements of artificial intelligence in image recognition, spacecraft modeling, and natural language processing are inspiring. Therefore, some researchers applied machine learning and deep learning methods to the classification of ECG signals. Hu et al. [9] extracted ECG characteristics through principal component analysis (PCA) and built a classifier with a support vector mechanism (SVM). Melgani et al. [10] applied the idea of particle swarm optimization to an SVM classifier and achieved a total accuracy of 89.72% for the classification of six arrhythmias. Kumar et al. [11] proposed the use of random forests (RF) to distinguish RR intervals and Park et al. [12] proposed a K-nearest neighbor (K-NN) classifier for detecting 17 types of ECG beats. Yuzhen et al. [13] attempted to use BP neural networks to classify ECG beats and achieved an accuracy rate of 93.9%. Ceylan et al. [14] used a feedforward neural network to detect four different arrhythmias with an average accuracy of 96.95%. With the application of the convolutional neural network (CNN) and recurrent neural network (RNN) in ECG classification, classification algorithms are becoming more and more mature. Kiranyaz et al. [15] designed a one-dimensional CNN to realize ECG classification by automatically extracting signal characteristics. Warrick et al. [16] combined CNN and long short-term memory (LSTM) to propose a new automatic detection and classification method for arrhythmias in ECG records. Übeyli et al. [17] used RNN with composite characteristics to detect ECG changes in some epilepsy patients. Shu et al. [18] used CNN and LSTM to categorize variable-length beat data from the MIT-BIT arrhythmia physio bank database, achieving an accuracy of 98.10% and sensitivity of 97.50% and specificity of 98.70%. Ji Y et al. [19] made a good overview of the existing methods and proposed a convolutional neural network based on faster regions, which converted one-dimensional ECG signals into two-dimensional images for five classifications with an average accuracy of 99.21%.

In this paper, 12-lead ECG records with richer information were used to design a deep learning algorithm to realize the intelligent classification of nine arrhythmias. Compared with previous studies, we constructed a deeper neural network with an effective attention mechanism introduced in this paper. Then, we systematically evaluated the influencing factors of the final classification results in more perspectives. Section 2 presents an overview of the related work. Data composition and processing are described in Section 3, followed by the detailed introduction of network construction and improvement in Section 4. After that, results and discussion are described in detail in Section 5. Lastly, conclusions are drawn in Section 6.

## 2. Related Work

The mainstream methods of using ECG signals to identify arrhythmia can be summarized into two categories: traditional signal processing techniques and classifiers constructed by neural networks. On the basis of filtering denoising and R point detection, the former method manually selects features and calculates statistical indicators, which has relatively low efficiency but strong interpretability; the latter method has similarities with neural network-based image recognition technology. Feature screening and parameter tuning are all done automatically by the network, which has relatively high efficiency but poor interpretability. The arrhythmia classification model based on machine learning can be seen as a transition from traditional methods to neural network methods.

Our work aims to establish an automatic classification system for arrhythmia from more heartbeats, more signal leads, and more arrhythmia categories, all while trying to explain the working mechanism of the proposed model from the perspective of a heat map.

## 3. Data

Deep learning models tend to train a large number of parameters, so having enough data is the basic premise. In this research, 14,000 12-lead ECG records from 3569 patients were used, which have

differences in noise size, sequence length, and amplitude range. Before designing the structure of a neural network, it is necessary to use some mathematical methods to process the data simply.

### 3.1. Data Description

In this work, half of the ECGs jointly tagged by several cardiologists were provided by Shanxi Bethune Hospital, which included a total of 7000 12-lead ECG records with a sampling rate of 500 Hz. They were divided into three parts, 5850 records as the training set, 650 records as the validation set, and 500 records as the testing set 1. Each record may have multiple different arrhythmia labels, so a record may be counted repeatedly in the respective statistics of different classes of signals. To make the results more convincing, we used 6500 ECG records collected offline from multiple channels by our teams as testing set two and 500 ECG records from the PhysioNet public database as testing set three [20,21]. The specific number of ECGs corresponding to nine types of arrhythmias contained in each data set is shown in Table 1. The sample segments of ECGs for nine categories are shown in Figure 1.

**Table 1.** Detailed composition of each data set. They all consist of the following nine types of ECG signals: no abnormality in ECG (Normal), atrial fibrillation (AF), first-degree atrioventricular block (FDAVB), complete right bundle branch block (CRBBB), left anterior fascicular block (LAFB), premature ventricular contraction (PVC), premature atrial contraction (PAC), early repolarization (ER), T wave change (TWC).

| Type | Trainset | Valset | Testset1 | Testset2 | Testset3 |
|---|---|---|---|---|---|
| **Normal** | 1752 | 202 | 173 | 1814 | 231 |
| **AF** | 455 | 47 | 41 | 506 | 41 |
| **FDAVB** | 479 | 58 | 42 | 571 | 45 |
| **CRBBB** | 738 | 87 | 66 | 842 | 50 |
| **LAFB** | 147 | 32 | 30 | 201 | 30 |
| **PVC** | 574 | 79 | 64 | 656 | 65 |
| **PAC** | 600 | 71 | 53 | 688 | 72 |
| **ER** | 196 | 18 | 11 | 317 | 12 |
| **TWC** | 1937 | 203 | 171 | 2273 | 164 |
| **Total** | 5850 | 650 | 500 | 6500 | 500 |

### 3.2. Data Processing

Clipping can make each ECG signal have the same length. The number of signal sampling points in the dataset range from 4500 to 50,000. Considering that the specific wave groups of PVC, PAC, and so on do not appear in all heartbeats, simply cutting the signal into a single or several heartbeats will lose the key information. After the overall analysis of the data, the signal length was cut to 8192 sampling points, including 16 or so cardiac beats. The ECG signal with insufficient length was filled with zero at the beginning. In this way, information integrity was guaranteed to the greatest extent, (compared with the irregularity of the waveform caused by the truncation of the single-heartbeat at an inappropriate position, the multi-heartbeat has information redundancy, thus the position of the starting point and ending point is no longer a restrictive factor) and unnecessary computational overhead was reduced. Compared with an entire ECG record that lasts for several hours, the information contained in 16 heartbeats is enough to reflect the essence. When training the neural network, each iteration adopted random clipping to make the data of the training set slightly different each time, which played a role of data enhancement to some extent.

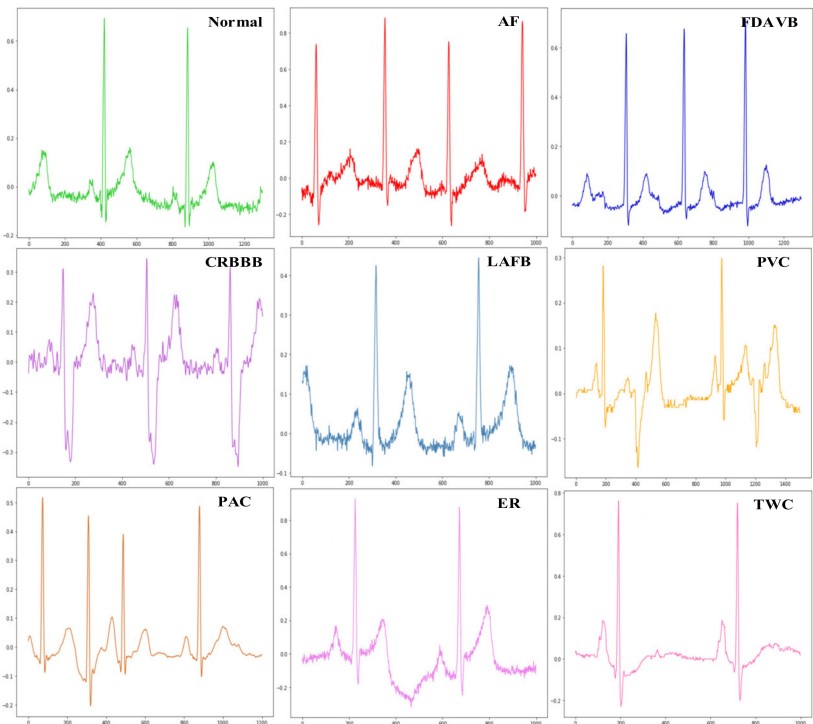

**Figure 1.** Nine types of arrhythmia waveform segments. It can be seen intuitively from the single-lead signal that the difference between categories is mainly reflected in wave frequency and wave morphology.

Normalization can convert data with different orders of magnitude into uniform measures. By calculating the population mean and population standard deviation, Z-Score standardization can shrink the signal amplitude to a smaller range according to the unified standard. In the derivative process of gradient descent, a small initial value will greatly accelerate the convergence rate. Its calculation formula is defined as:

$$v_{new} = \frac{v_{old} - \mu}{\delta} \tag{1}$$

Filtering is usually realized by the coordination of several bandpass filters with different cut-off frequencies. In traditional methods, filtering can effectively remove electromyographic signals, suppress power frequency interference, and remove baseline drift. However, the deep CNN can automatically eliminate noise interference in the feature extraction process, and the data quality of the dataset used is superior, so the effect of filtering on the final performance of the model is not obvious. In practice, we can control whether to add a filtering operation according to the specific experimental environment or to isolate and filter the signals with serious noise interference and several signals with high-quality requirements.

## 4. Material and Methods

The whole experiment was completed with the Spyder (Python 3.6) compiler based on the Python language. The construction of the neural network was implemented under the PyTorch (torch 1.2.0+cu92) framework, and the working platform was two GTX1080Ti graphics cards, an Intel Core i7 3.2 GHz (8700) processor, and a 32 GB RAM. In addition, NumPy, pandas, matplotlib, and other third-party libraries were used for data analysis and visualization.

CNN and RNN are two kinds of classical networks. The former can effectively process high-dimensional data by sharing parameters and selecting features automatically. The latter has a prominent advantage in the time series problem by virtue of its unique memorization. Inspired by GoogLeNet [22], a similar deep one-dimensional convolutional network is designed to compress information and obtain feature maps. The double-layer gated recurrent unit (GRU) structure after the

convolutional network is used to detect the sequence characteristics of the P-wave, QRS wave, T-wave, and other wave groups. The attention block is inserted at intervals in a cross-layer connection so that the inter-lead correlation is also fully considered. The network structure and data forward propagation are shown in Figure 2. The specific parameters of each layer of the network are given in Table 2.

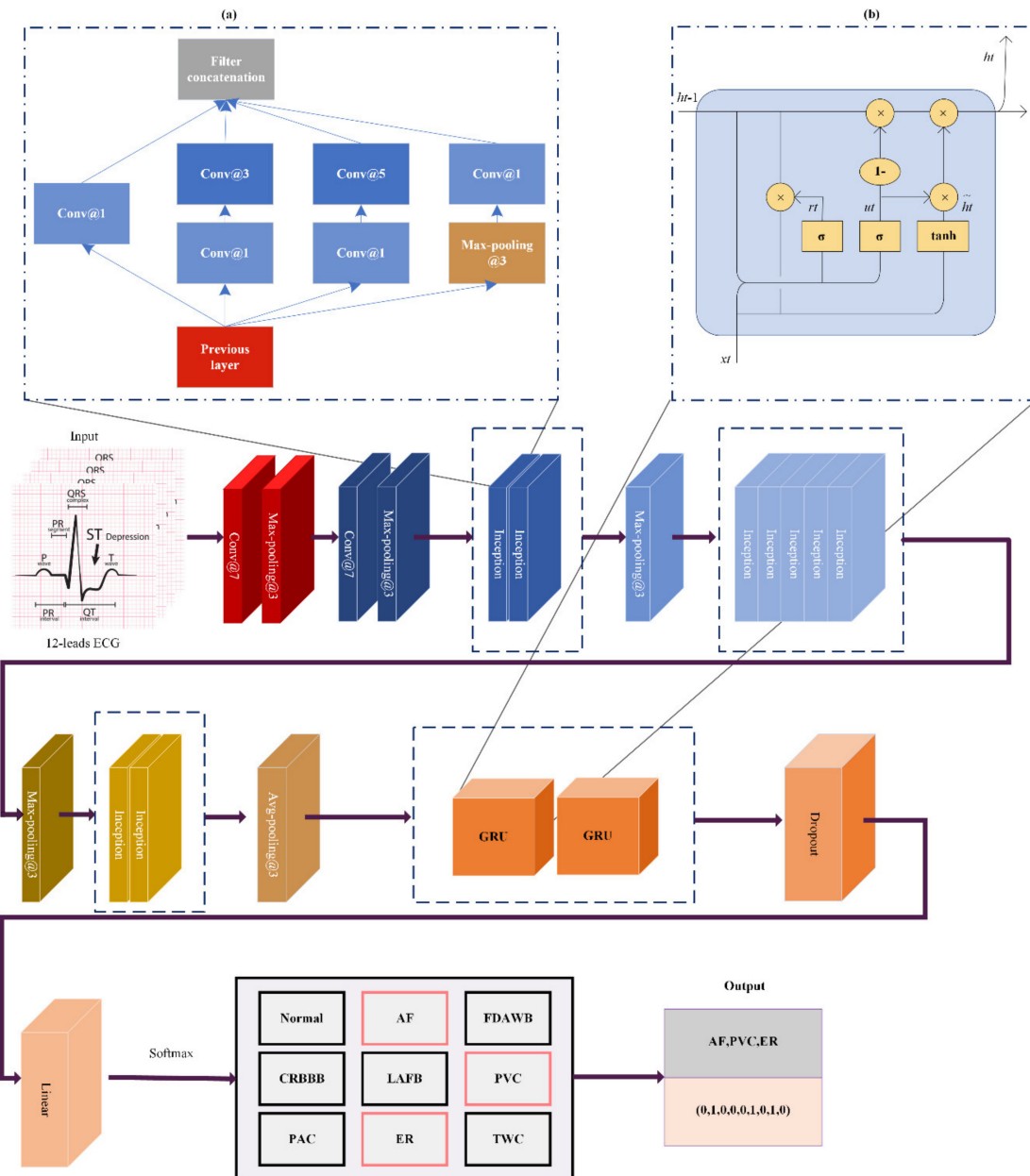

**Figure 2.** Architecture of the new neural network. (**a**) Typical Inception block. (**b**) Typical GRU structure. The input is a matrix of 12 × 8192. The basic components of the network include a one-dimensional convolutional layer, one-dimensional pooling layer, Inception, and GRU. A 1 × 9 output prediction vector is obtained by the softmax function.

**Table 2.** Detailed parameters of each layer. The relevant parameters of a total of 20 layers including the input layer are all given, and the number of output channels of each branch of Inception is separately listed.

| Layer | Type | Kernel Size/Stride | Output Size | Branch 1 | Branch 2 | Branch 3 | Branch 4 |
|---|---|---|---|---|---|---|---|
| 0 | Input | | $8192 \times 12$ | | | | |
| 1 | Convolution | $7 \times 1/2$ | $4096 \times 64$ | | | | |
| 2 | Max-pooling | $3 \times 1/2$ | $2048 \times 64$ | | | | |
| 3 | Convolution | $3 \times 1/1$ | $2048 \times 192$ | | | | |
| 4 | Max-pooling | $3 \times 1/2$ | $1024 \times 192$ | | | | |
| 5 | Inception | | $1024 \times 256$ | 64 | 128 | 32 | 32 |
| 6 | Inception | | $1024 \times 480$ | 128 | 192 | 96 | 64 |
| 7 | Max-pooling | $3 \times 1/2$ | $512 \times 480$ | | | | |
| 8 | Inception | | $512 \times 512$ | 192 | 208 | 48 | 64 |
| 9 | Inception | | $512 \times 512$ | 160 | 224 | 64 | 64 |
| 10 | Inception | | $512 \times 512$ | 128 | 256 | 64 | 64 |
| 11 | Inception | | $512 \times 528$ | 112 | 288 | 64 | 64 |
| 12 | Inception | | $512 \times 832$ | 256 | 320 | 128 | 128 |
| 13 | Max-pooling | $3 \times 1/2$ | $256 \times 832$ | | | | |
| 14 | Inception | | $256 \times 832$ | 256 | 320 | 128 | 128 |
| 15 | Inception | | $256 \times 1024$ | 384 | 384 | 128 | 128 |
| 16 | Avg-pooling | $3 \times 1/2$ | $128 \times 1024$ | | | | |
| 17 | GRU | | $128 \times 2048$ | | | | |
| 18 | GRU | | $128 \times 512$ | | | | |
| 19 | Linear | | $1 \times 9$ | Dropout (40%) | | Softmax | |

## 4.1. Inception Module

GoogLeNet's core architecture Inception extends the number of branches to four, improving overall network performance by increasing the depth and width of the network. In Figure 2a, one of the most classical structures of Inception is presented. A different size convolution kernel is adopted in each branch, which means different sizes of receptive fields can get different scale features. The addition of a $1 \times 1$ convolution kernel effectively reduces a large number of parameter computing bottlenecks caused by a large convolution kernel. Another significant advantage of Inception is that each of the convolution modules is actually composed of the convolution layer, BN layer, and ReLU activation layer. The introduction of batch normalization [23] has effectively solved the problem of gradient disappearance in deep networks and greatly accelerated the convergence rate of networks.

## 4.2. Attention Module

Combining the merits of Residual attention [24] and Dot-product attention [25], we designed a new attention module to work on the Max-pooling layer and Avg-pooling layer immediately after the Inception block. As shown in Figure 3, the initial 12-lead ECG signal gets the outline value of the Max-pooling layer by repeated convolution on the trunk with the increase of channels. The signal reaches the same length as the pooling layer by down-sampling on the branch with keeping the number of channels at 12, and then gets the Max-pooling layer detail value through the attention mechanism. Finally, the final value of the pooling layer is obtained by summing the branch value and the trunk value. The operation process can be expressed as the following equation:

$$Q_{new} = Q_{old} + softmax(Q_{old}K^T)K \tag{2}$$

where $Q_{new}$ is the feature map obtained after the input matrix $A$ flows through the attention module. $Q_{old}$ represents the output matrix of the pooling layer on the trunk, and $K$ represents the matrix directly obtained by the down-sampling on the branch. Therefore, $softmax(Q_{old}K^T)$ can be interpreted as the correlation coefficient matrix between the initial signal leads and the middle pooling layer channels. The feature map calculated by the attention mechanism contains both lead correlation information and wave shape information.

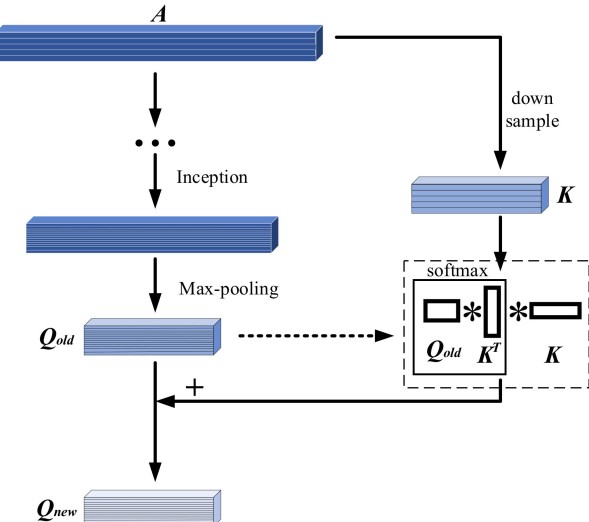

**Figure 3.** Attention mechanism action process. The output of Inception tends to increase significantly in the number of channels, so adding consideration to the initial 12 channels in pooling layers helps to pass on the correlation characteristics of inter-lead layer by layer.

### 4.3. GRU Module

GRU and LSTM are two good variants of RNN [26,27]. LSTM is used in most of the literature for the classification of ECG signals. However, in this experiment, after the CNN, we tried to add two layers of unidirectional GRU, unidirectional LSTM, bidirectional GRU, and bidirectional LSTM, respectively. The study found that GRU is better than LSTM, and unidirectional GRU is better than bidirectional GRU. The main difference between GRU and LSTM is that GRU uses one single gating unit to control both the decision of the forgetting factor and the decision of the update status. The update formula is as follows:

$$h_i^{(t)} = u_i^{(t-1)}h_i^{(t-1)} + (1 - u_i^{(t-1)})\sigma(b_i + \sum_j U_{i,j}x_j^{(t)} + \sum_j W_{i,j}r_j^{(t-1)}h_j^{(t-1)}) \tag{3}$$

where *u* stands for update gate and *r* stands for reset gate. $\sigma$ can be any kind of activation function. *b*, *U*, and *W* represent bias, input weights, and recurrent weights of the update gate in GRU cell, respectively. Inputting the update results of the hidden layer cells of the previous GRU layer to the hidden layer cells of the latter GRU layer can form a double-layer GRU, and inputting the last moment update results of the double-layer GRU into the full connection layer can complete the classification through the built-in Softmax function. The GRU structure used in this paper is shown in Figure 2b.

### 4.4. Other Important Components

In the experiment, Adam optimizer [28] was selected to complete the parameters update, and the initial learning rate was 0.0001. The 10-fold cross-validation was adopted to the train model, and the parameters with excellent prediction were selected to participate in the ensemble. Another key point is that a modified Binary Cross Entropy Loss was used as the loss function in this experiment rather than the commonly used MSE Loss or Cross Entropy Loss, which can be expressed as the following:

$$loss = \sum_i k_i\{-w[y \cdot \log x + (1 - y) \cdot \log(1 - x)]\} \tag{4}$$

where *x* is the network output vector mapped by Sigmoid function, namely the predicted value; *y* is the corresponding label vector, namely the true value; *w* represents the weights of loss calculation.

$k_i$ can be interpreted as the weights of categories to eliminate the impact of data asymmetry imbalance between classes.

## 5. Results and Discussion

In some of the literature, accuracy, sensitivity, and specificity were usually used as the final evaluation indexes of the model. In this experiment, there was a serious data imbalance, which can be reflected in the overall distribution of arrhythmias of nine categories to some extent. At first, we tried to eliminate the adverse effects by increasing or decreasing the number of various data to construct a balanced data set, but the results were not satisfactory. After that, the adjustment model and loss function were selected to fit the unbalanced data set. In this case, it was more reasonable to select precision rate, recall rate, and F1 score as evaluation indexes.

### 5.1. Results Analysis

The loss curve and F1 curve measured on the validation set are given in Figure 4. It can be seen that, after 80 iterations, in the case of the attention mechanism, loss is basically stable at 0.06, which is 0.02 lower than that without the attention mechanism. Meanwhile, the average F1 score of the network with attention mechanism is basically stable above 0.90, while that of the network without the attention mechanism is stable above 0.84. This suggests that the attention mechanism is very effective in learning the 12-lead ECG signals, improving by six percentage points compared with the model without the attention mechanism.

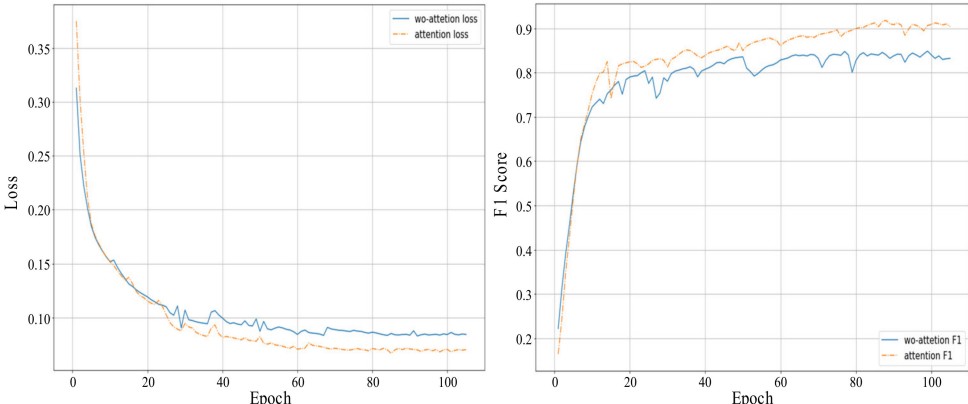

**Figure 4.** (**Left**) Loss curve. The blue line and yellow line, respectively, represent the decline curve of loss without the attention mechanism and with the attention mechanism. (**Right**) F1 curve. The blue line and yellow line, respectively, represent the rising curve of F1 score without the attention mechanism and with the attention mechanism.

On the basis of the attention mechanism, we added two groups of contrast experiments to systematically show that the initial signal length and the selection of RNN variants are important factors affecting the performance of the model. This aspect of the research was insufficient in most previous literature, Tables 3 and 4, respectively, give the results of the above mentioned two groups of contrast experiments on the validation set. As can be seen in Table 3, when the input signal length starts to decrease from 8192, F1 scores generally show a downward trend due to less and less cardiac beat information. When the input signal length increases upward from 8192, the original information cannot be highlighted effectively due to too much filling of zero, and the F1 score also shows a slight decline. The results in Table 4 reflect that GRU is better than LSTM, and a unidirectional network is better than a bidirectional network, which is contrary to the comparison results of RNN variants commonly used in text processing. We guess that in the deep structure used in combination with CNN, the signal length becomes shorter, but the number of channels increases significantly, and the unidirectional.

**Table 3.** Influence of input signals of different lengths on F1 scores. The 4096-length signal and 8192-length signal have their own advantages and disadvantages in single F1 scores, but the average F1 score shows that the length of 8192 is better. In addition, the main disadvantage of too long signal selection is the large amount of calculation. The main disadvantage of too short signal selection is serious information loss and F1 score drops significantly.

| Type | Length | | | | | |
|---|---|---|---|---|---|---|
| | **1024** | **2048** | **4096** | **8192** | **16,384** | **32,768** |
| Normal | 0.842 | 0.883 | 0.895 | 0.930 | 0.875 | 0.872 |
| AF | 0.931 | 0.968 | 0.971 | 0.968 | 0.953 | 0.953 |
| FDAVB | 0.754 | 0.864 | 0.877 | 0.920 | 0.852 | 0.844 |
| CRBBB | 0.937 | 0.971 | 1.000 | 0.983 | 0.973 | 0.962 |
| LAFB | 0.732 | 0.865 | 0.922 | 0.906 | 0.884 | 0.853 |
| PVC | 0.714 | 0.884 | 0.932 | 0.962 | 0.921 | 0.906 |
| PAC | 0.665 | 0.822 | 0.868 | 0.906 | 0.863 | 0.845 |
| ER | 0.381 | 0.684 | 0.684 | 0.757 | 0.702 | 0.702 |
| TWC | 0.704 | 0.852 | 0.881 | 0.904 | 0.875 | 0.841 |
| Average | 0.740 | 0.866 | 0.892 | 0.915 | 0.878 | 0.864 |

**Table 4.** Influence of different RNN variants on F1 scores. The four variants showed little difference in scores. Unidirectional GRU had a small advantage in terms of average F1 scores but bidirectional GRU showed its superiority in the single prediction of CRBBB, LAFB, and PAC.

| Type | Module | | | |
|---|---|---|---|---|
| | **Unidirectional GRU** | **Bidirectional GRU** | **Unidirectional LSTM** | **Bidirectional LSTM** |
| Normal | 0.930 | 0.891 | 0.891 | 0.874 |
| AF | 0.968 | 0.957 | 0.968 | 0.957 |
| FDAVB | 0.920 | 0.915 | 0.888 | 0.901 |
| CRBBB | 0.983 | 0.983 | 0.977 | 0.973 |
| LAFB | 0.906 | 0.922 | 0.906 | 0.884 |
| PVC | 0.962 | 0.949 | 0.943 | 0.943 |
| PAC | 0.906 | 0.915 | 0.901 | 0.904 |
| ER | 0.757 | 0.722 | 0.684 | 0.684 |
| TWC | 0.904 | 0.887 | 0.875 | 0.862 |
| Average | 0.915 | 0.905 | 0.893 | 0.887 |

GRU, with the simplest structure, is easier to find the point of the local minimum.

## 5.2. Model Evaluation

The confusion matrix drawn according to the best classification result of the validation set is shown in Figure 5. Most samples of abnormal distribution are explicable. For example, the large number of false-positive samples of normal type is because all empty prediction results below the defined threshold are classified into normal artificially. TWC is confused with the other eight categories to varying degrees, which is partly caused by the elimination of small fluctuations in the convolution process and the serious drift of the initial sampling points. In addition, it can be seen that the negative effects of false positive and false negative samples are amplified due to the few ECG records of ER in the data set. From the perspective of morphology, the model cannot effectively learn the feature of J point elevation, which also restricts the prediction score of ER type.

From the perspective of the confusion matrix, the learning effect of the model is outstanding. In order to have an objective and comprehensive understanding of the model, we calculated precision rate, recall rate, specificity, and F1 score of all categories on the validation set and three testing sets, which were summarized in Table 5. The highest average F1 score was 0.919 and the lowest was 0.886. It can be clearly seen that AF, FDAVB, CRBBB, PVC, and PAC have stable performance in four data sets and good generalization ability. When the number of testing samples increases to 6500 (set3), the inter-class confusion between normal and TWC is exacerbated. There are over-fitting risks because

of the extremely limited number of LAFB and ER in the training set. Therefore, the F1 scores of these two categories will also decline on a more diverse testing set.

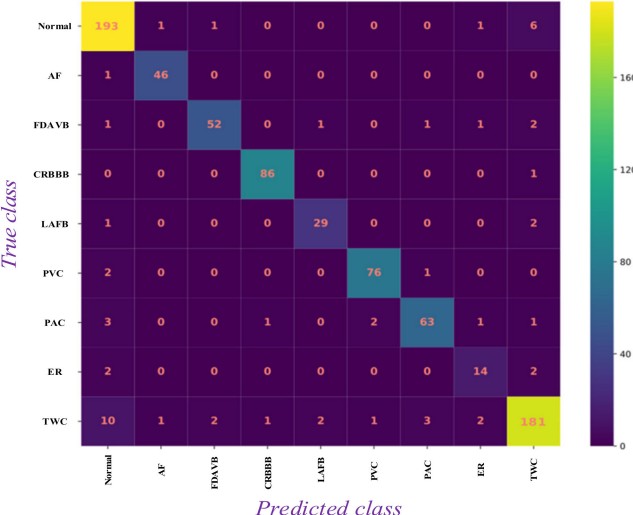

**Figure 5.** Confusion matrix on the validation set. The samples on the main diagonal occupied the majority and a small number of mistakes mainly occurred in the surrounding areas. There was a great correlation between inter-class confusion and inter-class morphological similarity.

**Table 5.** Comprehensive performance of the model on different data sets. Set1, Set2, Set3, and Set4 correspond to Valset, Testset1, Testset2, and Testset3 of Table 1, respectively.

| | Precision (PPV) | | | | Recall (Sensitivity) | | | | Specificity | | | | F1 Score | | | |
|---|---|---|---|---|---|---|---|---|---|---|---|---|---|---|---|---|
| | Set1 | Set2 | Set3 | Set4 | Set1 | Set2 | Set3 | Set4 | Set1 | Set2 | Set3 | Set4 | Set1 | Set2 | Set3 | Set4 |
| Normal | 0.906 | −0.921 | 0.861 | 0.883 | 0.955 | −0.949 | 0.914 | 0.935 | 0.955 | −0.955 | 0.942 | 0.951 | 0.930 | 0.935 | 0.887 | 0.908 |
| AF | 0.958 | 0.979 | 0.962 | 0.976 | 0.978 | 0.987 | 0.983 | 1.000 | 0.995 | 0.997 | 0.996 | 0.997 | 0.968 | 0.983 | 0.972 | 0.987 |
| FDAVB | 0.945 | 0.939 | 0.933 | 0.933 | 0.897 | 0.886 | 0.891 | 0.889 | 0.996 | 0.995 | 0.995 | 0.995 | 0.920 | 0.912 | 0.912 | 0.910 |
| CRBBB | 0.977 | 1.000 | 0.975 | 0.980 | 0.989 | 1.000 | 0.988 | 0.980 | 0.995 | 1.000 | 0.996 | 0.996 | 0.983 | 1.000 | 0.981 | 0.980 |
| LAFB | 0.906 | 0.900 | 0.872 | 0.900 | 0.906 | 0.871 | 0.838 | 0.844 | 0.995 | 0.993 | 0.987 | 0.991 | 0.906 | 0.885 | 0.852 | 0.871 |
| PVC | 0.962 | 0.974 | 0.967 | 0.969 | 0.962 | 0.974 | 0.956 | 0.969 | 0.995 | 0.996 | 0.995 | 0.995 | 0.962 | 0.974 | 0.961 | 0.969 |
| PAC | 0.926 | 0.909 | 0.914 | 0.917 | 0.887 | 0.896 | 0.879 | 0.889 | 0.991 | 0.992 | 0.990 | 0.991 | 0.906 | 0.902 | 0.896 | 0.903 |
| ER | 0.737 | 0.750 | 0.672 | 0.692 | 0.778 | 0.818 | 0.688 | 0.750 | 0.992 | 0.992 | 0.974 | 0.983 | 0.757 | 0.783 | 0.680 | 0.720 |
| TWC | 0.914 | 0.906 | 0.841 | 0.884 | 0.892 | 0.884 | 0.823 | 0.854 | 0.969 | 0.961 | 0.953 | 0.959 | 0.904 | 0.895 | 0.832 | 0.869 |
| Average | 0.914 | 0.920 | 0.889 | 0.904 | 0.916 | 0.918 | 0.884 | 0.901 | 0.987 | 0.987 | 0.981 | 0.984 | 0.915 | 0.919 | 0.886 | 0.902 |

### 5.3. Comparison with Other Approaches

There are endless algorithms for the classification of arrhythmias based on ECG signals, so we selected several representative machine learning methods and deep learning methods for experiments. Table 6 shows the comparison results with previous work. In terms of accuracy, sensitivity, and specificity, our model performs well, with specificity being the highest score. Compared with the method in literature 19, the model in this paper is slightly lower in accuracy and sensitivity. We believe that the use of Faster R-CNN can accurately locate a single heartbeat, but the network proposed by us is lacking in the discrimination of the wave groups position relationship with the simultaneous input of multiple heartbeats. Figure 6 shows the situation of classification errors. Convolution and pooling operations will shorten the distance of the wave groups and reduce the amplitude of the wave groups, which will eventually lead to misjudgment.

**Table 6.** In order to eliminate the influence of data sets on the diagnosis results, all the following models were trained on the trainset and uniformly tested and evaluated on the testset3.

| Author | Method | Acc (%) | Sen (%) | Spe (%) |
|---|---|---|---|---|
| Proposed | Inception + GRU + Attention | 92.84 | 90.12 | 98.41 |
| Melgani et al. [10] | SVM | 85.72 | 84.29 | 89.61 |
| Kumar et al. [11] | Random Forest | 87.14 | 87.23 | 91.97 |
| Oh, S.L. et al. [18] | CNN + LSTM | 92.10 | 90.50 | 97.42 |
| Ji Y et al. [19] | Faster R-CNN | 93.25 | 91.16 | 98.23 |
| Zubair et al. [29] | CNN | 90.40 | 87.75 | 94.42 |
| Prasad et al. [30] | K-NN | 86.45 | 85.11 | 91.36 |
| Han et al. [31] | Residual Block | 91.19 | 89.96 | 96.56 |

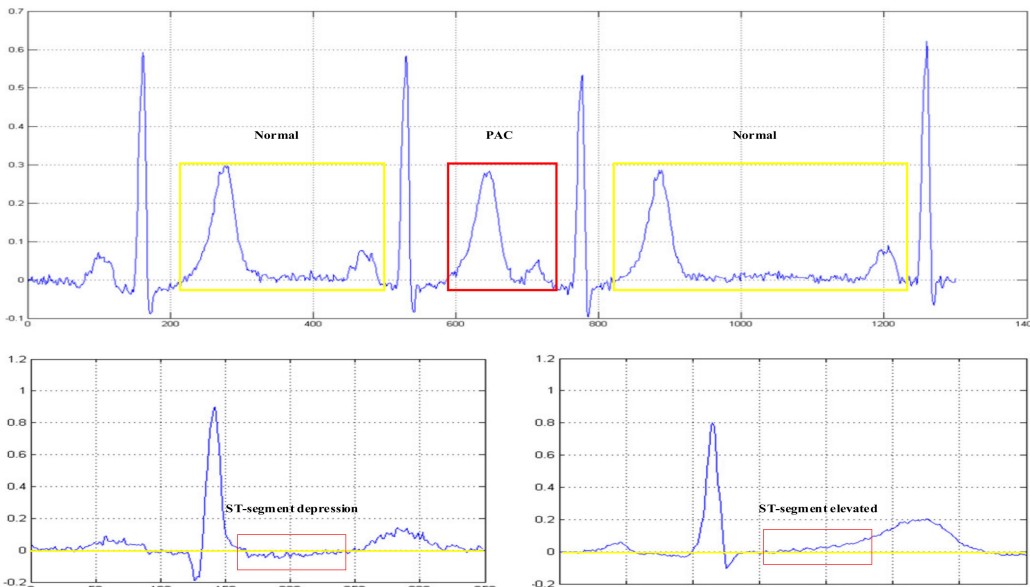

**Figure 6.** Examples of misclassifications. The advance of the P wave and the amplitude of the ST segment are important for the basis of classification.

Based on the consideration of clinical interpretability, we obtained the summary of the key regions shown in Figure 7 through Grad-CAM, which reflects that the concerns of the classifier are basically the same as the concerns of the professional physicians.

*5.4. Future Work*

In view of some problems existing in the model, future research mainly includes the following three aspects. Firstly, the improvement of data processing methods and optimization of network structure are necessary, with the hope of achieving similar results after reducing the network size and parameter training volume. Secondly, trying to combine neural networks with PCA, RF, and K-NN of machine learning to build a new model to realize accurate identification of ER class. The biggest difficulty is to find out the subtle changes between QRS and T waves and extract these morphological features effectively. Thirdly, changing the ensemble strategy. When multiple labels are attached to an ECG record, effective logical evaluation can eliminate impossible combinations of labels. For the null prediction results, we can return them to the model for the second fine division. In addition, ECG records are often accompanied by some personal information such as the patient's gender and age, which can be fully utilized to assist judgment in the ensemble process.

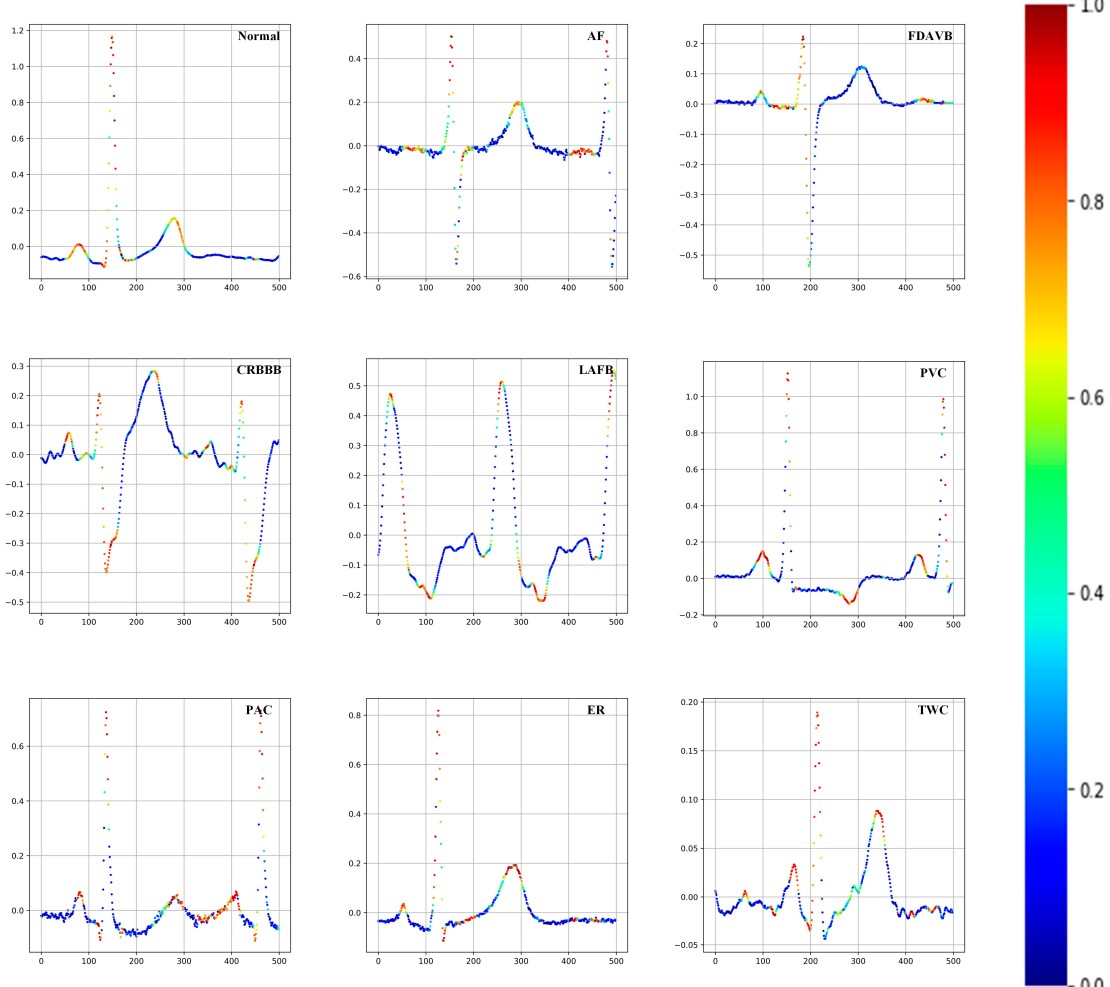

**Figure 7.** The heat curves of arrhythmias. Warm-toned points have more influence than cool-toned points in model decision making.

## 6. Conclusions

The biggest innovation of this research was to propose a deep neural network model with attention mechanism, which can directly classify the 12-lead ECG records. The importance of signal initial length selection and RNN variant selection were also illustrated by contrast experiments. According to multiple testing sets, the F1 score was stable at 0.886 and up to 0.919. The F1 scores of AF, CRBBB, and PVC all reached above 0.96 and showed stable performance, which had considerable clinical application value. ER and TWC still had shortcomings in recognition but some constructive ideas for improvement had been proposed. The heat curve mentioned at the end of the article has long-term significance for the automatic classification system of arrhythmias to be widely recognized by clinicians. At the same time, the fixed length of the signal, the fixed number of leads, and the fixed number of categories will make the model lack flexibility, which is the main limitation.

**Author Contributions:** D.L. and H.W. conceived and designed the experiments; H.W. performed the experiments; Y.T. and J.F. analyzed the data; D.L. contributed analysis tools; J.Z. wrote the paper. All authors have read and agreed to the published version of the manuscript.

**Funding:** This research received no external funding. The paper supported by The General Object of National Natural Science Foundation (62076177) Study on the risk Assessment Model of heart failure by integrating multi-modal big data; The General Object of National Natural Science Foundation (61772358) Research on the key technology of BDS precision positioning in complex landform; Project supported by International Cooperation Project of Shanxi Province (Grant No. 201603D421014) Three-Dimensional Reconstruction Research of Quantitative Multiple Sclerosis Demyelination; International Cooperation Project of Shanxi Province (Grant No.

201603D421012): Research on the key technology of GNSS area strengthen information extraction based on crowd sensing; Shanxi Province key core technology and generic technology research and development special (project No. 2020XXX007) Energy Internet Integrated Intelligent Data Management and Decision Support Platform.

**Conflicts of Interest:** The authors declare no conflict of interest.

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
