# Peer review of "Automatic Classification System of Arrhythmias Using 12-Lead ECGs with a Deep Neural Network Based on an Attention Mechanismâ€"

_symmetry, doi:10.3390/sym12111827_

Round 1
Reviewer 1 Report
Please explain why did you chose a total number of 20 layers.
The scientific background of the paper can be improved.
Please consider reorganizing the Abstract in a compact form.
Reviewer 2 Report
The paper proposes a deep learning-based solution for implementing an intelligent classifier of arrhythmias.
Although the submission addresses a very interesting research topic, which is relevant for the journal, the paper presents several weaknesses, which call for a major:
- The paper is not well written, there are some repeated sentences, and typos. Please, fix them.
- In Section 1.Introduction, pag 3 the sentence is not compressible “Compared with previous studies, we constructed a deeper neural network, introduced an effective attention mechanism and systematically evaluated the influencing factors of the final classification results on more subjects.” Please, rephrase it.
- In Section 2.2 Data Processing, it is not clear how the integrity is guaranteed.
- There is confusion on the features employed as input to the proposal. This point should be better clarified. I suggest to add a table reporting the features used, their typology and dimension.
- A Background Section describing the approaches employed to build the proposal should be added, in order to make more readable the manuscript.
- The introduction Section should report a brief introduction related to the potentialities of Machine Learning in solving tasks in numerous academic and industrial fields. In this regard, the authors are suggested to consult and cite in the manuscript the followings:
“Malware detection in mobile environments based on Autoencoders and API-images”, Journal of Parallel and Distributed Computing, 2020, 137, pp. 26-33, doi: 10.1016/j.jpdc.2019.11.001
"Spacecraft autonomy modeled via Markov decision process and associative rule-based machine learning", 2017 IEEE International Workshop on Metrology for AeroSpace (MetroAeroSpace), Padua, 2017, pp. 324-329, doi: 10.1109/MetroAeroSpace.2017.7999589.
- The mathematical terminology need to be reviewed.
- Usually, the experiments involving Machine Learning approaches are performed by dividing the dataset in training and testing set. Then, the training is, in turn, divided in learning and evaluation. These two subsets are employed for adjusting parameters. This is not explicitly described in the paper. Details should be given.
- Although I am in agreement with the authors about the choice of the F1 measure to be used for the model evaluation, it is necessary to report also the confusion matrix of the testing dataset.
- The conclusion needs to be extended. Besides, the main limitations of the proposal should be reported in the conclusions.
Reviewer 3 Report
Authors showed the automatic classification system of arrhythmias using 12-leads ECG with DL. The accuracy is good. This paper is well written. There are some issues in this study.
1) The clinical implication is unknown. In Table 6, Ref #14 showed the accuracy is superior to your model. 12-leads ECG model's advantage is limited. You should discuss about this.
2) Explainable is important in this field. Please use the other method to explain your results (eg. GrandCAM).
3) Examples of misclassified cases should be shown.
Round 2
Reviewer 2 Report
The authors have successfully addressed any of my questions. From a technical point of view, the manuscript in its current form is ready to be published.
Reviewer 3 Report
No further comment.